# COVID-19 Public Sentiment Insights and Machine Learning for Tweets Classification

**Jim Samuel** [1,*], **G. G. Md. Nawaz Ali** [2,*], **Md. Mokhlesur Rahman** [3,4], **Ek Esawi** [5] **and Yana Samuel** [6]

1. Department of Business Analytics, University of Charleston, Charleston, WV 25304, USA
2. Department of Applied Computer Science, University of Charleston, Charleston, WV 25304, USA
3. The William States Lee College of Engineering, University of North Carolina at Charlotte, Charlotte, NC 28223, USA; mrahma12@uncc.edu
4. Department of Urban and Regional Planning (URP), Khulna University of Engineering & Technology (KUET), Khulna 9203, Bangladesh
5. Department of Data Analytics, University of Charleston, Charleston, WV 25304, USA; eesawi@ucwv.edu
6. Department of Education, Northeastern University, Boston, MA 02115, USA; yana.samuel@gmail.com
* Correspondence: jim@aiknowledgecenter.com (J.S.); ggmdnawazali@ucwv.edu (G.G.M.N.A.)

**Abstract:** Along with the Coronavirus pandemic, another crisis has manifested itself in the form of mass fear and panic phenomena, fueled by incomplete and often inaccurate information. There is therefore a tremendous need to address and better understand COVID-19's informational crisis and gauge public sentiment, so that appropriate messaging and policy decisions can be implemented. In this research article, we identify public sentiment associated with the pandemic using Coronavirus specific Tweets and R statistical software, along with its sentiment analysis packages. We demonstrate insights into the progress of fear-sentiment over time as COVID-19 approached peak levels in the United States, using descriptive textual analytics supported by necessary textual data visualizations. Furthermore, we provide a methodological overview of two essential machine learning (ML) classification methods, in the context of textual analytics, and compare their effectiveness in classifying Coronavirus Tweets of varying lengths. We observe a strong classification accuracy of 91% for short Tweets, with the Naïve Bayes method. We also observe that the logistic regression classification method provides a reasonable accuracy of 74% with shorter Tweets, and both methods showed relatively weaker performance for longer Tweets. This research provides insights into Coronavirus fear sentiment progression, and outlines associated methods, implications, limitations and opportunities.

**Keywords:** COVID-19; Coronavirus; machine learning; sentiment analysis; textual analytics; twitter

## 1. Introduction

In this research article, we cover four critical issues: (1) public sentiment associated with the progress of Coronavirus and COVID-19, (2) the use of Twitter data, namely Tweets, for sentiment analysis, (3) descriptive textual analytics and textual data visualization, and (4) comparison of textual classification mechanisms used in artificial intelligence (AI). The rapid spread of Coronavirus and COVID-19 infections have created a strong need for discovering efficient analytics methods for understanding the flow of information and the development of mass sentiment in pandemic scenarios. While there are numerous initiatives analyzing healthcare, preventative, care and recovery, economic and network data, there has been relatively little emphasis on the analysis of aggregate personal level and social media communications. McKinsey [1] recently identified critical aspects

for COVID-19 management and economic recovery scenarios. In their industry-oriented report, they emphasized data management, tracking and informational dashboards as critical components of managing a wide range of COVID-19 scenarios.

There has been an exponential growth in the use of textual analytics, natural language processing (NLP) and other artificial intelligence techniques in research and in the development of applications. Despite rapid advances in NLP, issues surrounding the limitations of these methods in deciphering intrinsic meaning in text remain. Researchers at CSAIL, MIT (Computer Science and Artificial Intelligence Laboratory, Massachusetts Institute of Technology), demonstrated how even the most recent NLP mechanisms can fall short and thus remain "vulnerable to adversarial text" [2]. It is, therefore, important to understand inherent limitations of text classification techniques and relevant machine learning algorithms. Furthermore, it is important to explore whether multiple exploratory, descriptive and classification techniques contain complimentary synergies which will allow us to leverage the "whole is greater than the sum of its parts" principle in our pursuit for artificial intelligence driven insights generation from human communications. Studies in electronic markets demonstrated the effectiveness of machine learning in modeling human behavior under complex informational conditions, highlighting the role of the nature of information in affecting human behavior [3]. The source data for all Tweets data analysis, tables and every figure, including the fear curve in Figure 1 below, in this research consists of publicly available Tweets data, specifically downloaded for the purposes of this research and further described in the Data acquisition and preparation Section 3.1.1 of this study.

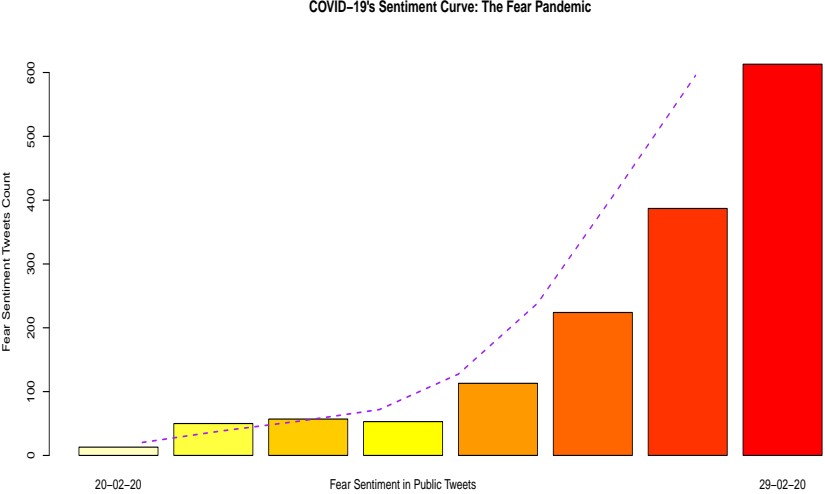

**Figure 1.** Fear curve.

The rise in emphasis on AI methods for textual analytics and NLP followed the tremendous increase in public reliance on social media (e.g., Twitter, Facebook, Instagram, blogging, and LinkedIn) for information, rather than on the traditional news agencies [4–6]. People express their opinions, moods, and activities on social media about diverse social phenomena (e.g., health, natural hazards, cultural dynamics, and social trends) due to personal connectivity, network effects, limited costs and easy access. Many companies are using social media to promote their product and service to the end-users [7]. Correspondingly, users share their experiences and reviews, creating a rich reservoir of information stored as text. Consequently, social media and open communication platforms are becoming important sources of information for conducting research, in the contexts of rapid development of information and communication technology [8]. Researchers and practitioners mine massive textual and unstructured datasets to generate insights about mass behavior, thoughts and emotions on a wide variety of issues such as product reviews, political opinions and trends,

motivational principles and stock market sentiment [4,9–13]. Textual data visualization is also used to identify the critical trend of change in fear-sentiment, using the "Fear Curve" in Figure 1, with the dotted Lowess line demonstrating the trend, and the bars indicating the day to day increase in fear Tweets count. Tweets were first classified using sentiment analysis, and then the progression of the fear-sentiment was studied, as it was the most dominant emotion across the entire Tweets data. This exploratory analysis revealed the significant daily increase in fear-sentiment towards the end of March 2020, as shown in Figure 1.

In this research article, we present textual analyses of Twitter data to identify public sentiment, specifically, tracking the progress of fear, which has been associated with the rapid spread of Coronavirus and COVID-19 infections. This research outlines a methodological approach to analyzing Twitter data specifically for identification of sentiment, key words associations and trends for crisis scenarios akin to the current COVID-19 phenomena. We initiate the discussion and search for insights with descriptive textual analytics and data visualization, such as exploratory Word Clouds and sentiment maps in Figures 2–4.

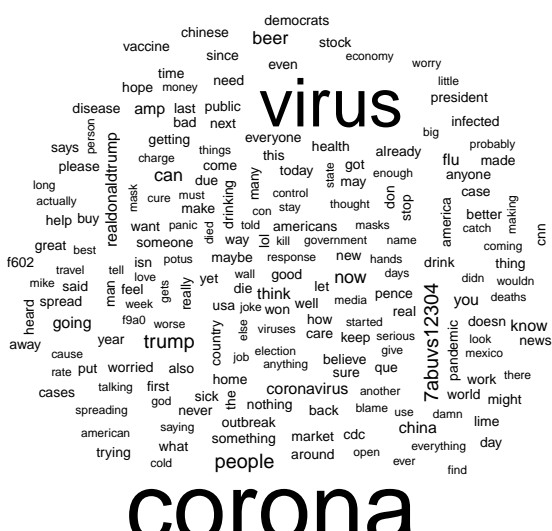

**Figure 2.** An instance of word cloud in twitter data.

Early stage exploratory analytics of Tweets revealed interesting aspects, such as the relatively higher number of Coronavirus Tweets coming from iPhone users, as compared to Android users, along with a proportionally higher use of word-associations with politics (mention of Republican and Democratic party leaders), URLs and humour, depicted by the word-association of beer with Coronavirus, as summarized in Table 1 below. We observed that such references to humour and beer were overtaken by "Fear Sentiment" as COVID-19 progressed and its seriousness became evident (Figure 1). Tweets insights with textual analytics and NLP thus serve as a good reflector of shifts in public sentiment.

**Table 1.** Tweet features summarized by source category.

| Source | Total | Hashtags | Mentions | Urls | Pols | Corona | Flu | Beer | AbuseW |
|---|---|---|---|---|---|---|---|---|---|
| iPhone | 3281 | 495 | 2305 | 77 | 218 | 4238 | 171 | 336 | 111 |
| Android | 1180 | 149 | 1397 | 37 | 125 | 1050 | 67 | 140 | 41 |
| iPad | 75 | 6 | 96 | 4 | 12 | 85 | 4 | 8 | 2 |
| Cities | 30 | 0 | 0 | 0 | 0 | 0 | 0 | 0 | 0 |

One of the key contributions of this research is our discussion, demonstration and comparison of Naïve Bayes and Logistic methods-based textual classification mechanisms commonly used in AI

applications for NLP, and specifically contextualized in this research using machine learning for Tweets classifications. Accuracy is measured by the ratio of correct classifications to the total number of test items. We observed that Naïve Bayes is better for small to medium size tweets and can be used for classifying short Coronavirus Tweets sentiments with an accuracy of 91%, as compared to logistic regression with an accuracy of 74%. For longer Tweets, Naïve Bayes provided an accuracy of 57% and logistic regression provided an accuracy of 52%, as summarized in Tables 6 and 7.

## 2. Literature Review

This study was informed by research articles from multiple disciplines and therefore, in this section, we cover literature review of textual analytics, sentiment analysis, Twitter and NLP, and machine learning methods. Machine learning and strategic structuring of information characteristics are necessary to address evolving behavioral issues in big data [3]. Textual analytics deals with the analysis and evocation of characters, syntactic features, semantics, sentiment and visual representations of text, its characteristics, and associated endogenous and exogenous features. Endogenous features refer to aspects of the text itself, such as the length of characters in a social media post, use of keywords, use of special characters and the presence or absence of URL links and hashtags, as illustrated for this study in Table 2. These tables summarize the appearances of "mentions" and "hashtags" in descending order, indicating the use of screen names and "#" symbol within the text of the Tweet, respectively.

**Table 2.** Summary of endogenous features.

| Tagged | Frequency | Hashtag | Frequency |
|---|---|---|---|
| realDonaldTrump | 74 | coronavirus | 23 |
| CNN | 21 | DemDebate | 16 |
| ImtiazMadmood | 16 | corona | 8 |
| corona | 13 | CoronavirusOutbreak | 8 |
| AOC | 12 | CoronaVirusUpdates | 7 |
| coronaextrausa | 12 | coronavirususa | 7 |
| POTUS | 12 | Corona | 6 |
| CNN MSNBC | 11 | COVID19 | 5 |

Exogenous variables, in contrast, are those aspects which are external but related to the text, such as the source device used for making a post on social media, location of Twitter user and source types, as illustrated for this study in Table 3. The Table summarizes "source device" and "screen names", indicating variables representing type of device used post the Tweet, and the screen name of the Twitter user, respectively, both external to the text of the Tweet. Such exploratory summaries describe the data succinctly, provide a better understanding of the data, and helps generate insights which inform subsequent classification analysis. Past studies explored custom approaches to identifying constructs such as dominance behavior in electronic chat, indicating the tremendous potential for extending such analyses by using machine learning techniques to accelerate automated sentiment classification and the subsections that follow present key insights gained from literature review to support and inform the textual analytics processes used in this study [14–17].

**Table 3.** Summary of exogenous features.

| Source | Frequency | Screen Name | Frequency |
|---|---|---|---|
| Twitter for iPhone | 3281 | _CoronaCA | 30 |
| Twitter for Android | 1180 | MBilalY | 25 |
| Twitter for iPad | 75 | joanna_corona | 17 |
| Cities | 30 | eads_john | 13 |
| Tweetbot for i<U+039F>S | 29 | _jvm2222 | 11 |
| CareerArc2.0 | 14 | AlAboutNothing | 11 |
| Twitter Web Client | 16 | dallasreese | 9 |
| 511NY-Tweets | 3 | CpaCarter | 8 |

## 2.1. Textual Analytics

A diverse array of methods and tools were used for textual analytics, subject to the nature of the textual data, research objectives, size of dataset and context. Twitter data has been used widely for textual and emotions analysis [18–20]. In another instance, a study analyzing customer feedback for a French Energy Company using more than 70,000 tweets published over a year [21], used a Latent Dirichlet Allocation algorithm to retrieve interesting insights about the energy company, hidden due to data volume, by frequency-based filtering techniques. Poisson and negative binomial models were used to explore Tweet popularity as well. The same study also evaluated the relationship between topics using seven dissimilarity measures and found that Kullback-Leibler and the Euclidean distances performed better in identifying related topics useful for user-based interactive approach. Similarly, extant research applying Time Aware Knowledge Extraction (TAKE) methodology [22] demonstrated methods to discover valuable information from huge amounts of information posted on Facebook and Twitter. The study used topic-based summarizing of Twitter data to explore content of research interest. Similarly, they applied a framework which uses less detailed summary to produce good quality information. Past research has also investigated the usefulness of twitter data to assess personality of users, using DISC (Dominance, Influence, Compliance and Steadiness) assessment techniques [23]. Similar research has been used in information systems using textual analytics to develop designs for identification of human traits, including dominance in electronic communication [17]. DISC assessment is useful for information retrieval, content selection, product positioning and psychological assessment of users. So also, a combination of psychological and linguistic analysis was used in past research to extract emotions from multilingual text posted on social media [24].

## 2.2. Twitter Analytics

Extant research has evaluated the usefulness of social media data in revealing situational awareness during crisis scenarios, such as by analyzing wildfire-related Twitter activities in San Diego County, modeling with about 41,545 wildfire related tweets, from May of 2014, [11]. Analysis of such data showed that six of the nine wildfires occurred on May 14, associated with a sudden increase of wildfire tweets on May 14. Kernel density estimation showed the largest hot spots of tweets containing "fire" and "wildfire" were in the downtown area of San Diego, despite being far away from the fire locations. This shows a geographical disassociation between fact and Tweet. Analysis of Twitter data in the current research also showed some disassociation between Coronavirus Tweets sentiment and actual Coronavirus hot spots, as shown in Figure 3. Such disassociation can be explained to some extent by the fact that people in urban areas have better access to information and communication technologies, resulting in a higher number of tweets from urban areas. The same study on San Diego wildfires also found that a large number of people tweeted "evacuation", which presented a useful cue about the impact of the wildfire. Tweets also demonstrated emphasis on wildfire damage (e.g., containment percentage and burnt acres) and appreciation for firefighters. Tweets, in the wildfire scenario, enhanced situational awareness and accelerated disaster response activities. Social network

analysis demonstrated that elite users (e.g., local authorities, traditional media reporters) play an important role in information dissemination and dominated the wildfire retweet network.

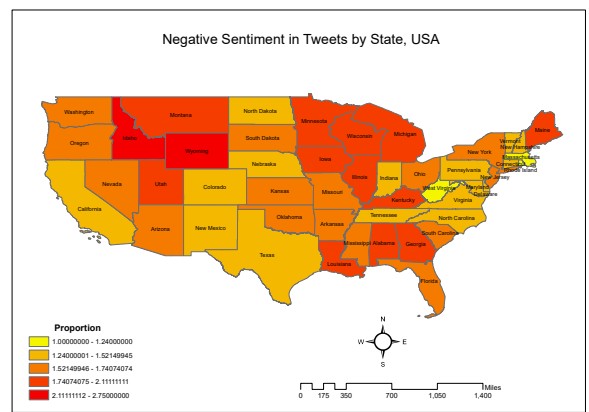

(**a**) Negative sentiment.

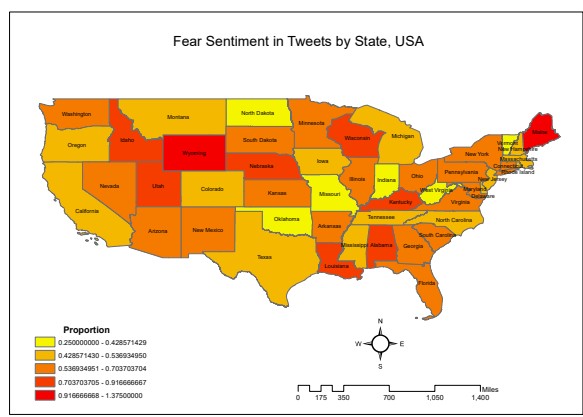

(**b**) Fear sentiment.

**Figure 3.** Sentiment map.

Twitter data has also been extensively used for crisis situations analysis and tracking, including the analysis of pandemics [25–28]. Nagar et al. [29] validated the temporal predictive strength of daily Twitter data for influenza-like illness for emergency department (ILI-ED) visits during the New York City 2012–2013 influenza season. Widener and Li (2014) [8] performed sentiment analysis to understand how geographically located tweets on healthy and unhealthy food are geographically distributed across the US. The spatial distribution of the tweets analyzed showed that people living in urban and suburban areas tweet more than people living in rural areas. Similarly, per capita food tweets were higher in large urban areas than in small urban areas. Logistic regression revealed that tweets in low-income areas were associated with unhealthy food related Tweet content. Twitter data has also been used in the context of healthcare sentiment analytics. De Choudhury et al. (2013) [10] investigated behavioral changes and moods of new mothers in the postnatal situation. Using Twitter posts this study evaluated postnatal changes (e.g., social engagement, emotion, social network, and linguistic style) to show that Twitter data can be very effective in identifying mothers at risk of postnatal depression. Novel analytical frameworks have also been used to analyze supply chain management (SCM) related twitter data about, providing important insights to improve SCM practices and research [30]. They conducted descriptive analytics, content analysis integrating text mining and sentiment analysis, and network analytics on 22,399 SCM tweets. Carvaho et al. [31] presented an efficient platform named MISNIS (intelligent Mining of Public Social Networks' Influence in Society) to collect, store, manage, mine and visualize Twitter and Twitter user data. This platform allows non-technical users to mine data easily and has one of the highest success rates in capturing flowing Portuguese language tweets.

### 2.3. Classification Methods

Extant research has used diverse textual classification methods to evaluate social media sentiment. These classifiers are grouped into numerous categories based on their similarities. The section that follows discusses details about four essential classifiers we reviewed, including linear regression and K-nearest neighbor, and focuses on the two classifiers we chose to compare, namely Naïve Bayes and logistic regression, their main concepts, strengths, and weaknesses. The focus of this research is to present a machine learning-based perspective on the effectiveness of the commonly used Naïve Bayes and logistic regression methods.

2.3.1. Linear Regression Model

Although linear regression is primarily used to predict relationships between continuous variables, linear classifiers can also used to classify texts and documents [32]. The most common estimation method using linear classifiers is the least squares algorithm which minimizes an objective function (i.e., squared difference between the predicted outcomes and true classes). The least squares algorithm is similar to maximum likelihood estimation when outcome variables are influenced by Gaussian noise [33]. Linear ridge regression classifier optimizes the objective function by adding a penalizer to it. Ridge classifier converts binary outcomes to $-1, 1$ and treats the problem as a regression (multi-class regression for a multi-class problem) [34].

2.3.2. Naïve Bayes Classifier

Naïve Bayes classifier (NBC) is a proven, simple and effective method for text classification [35]. It has been used widely for document classification since the 1950s [36]. This classifier is theoretically based on the Bayes theorem [32,34,37]. A discussion on the mathematical formulation of NBC from a textual analytics perspective is provided under the methods section. NBC uses maximum a posteriori estimation to find out the class (i.e., features are assigned to a class based on the highest conditional probability). There are mainly two models of NBC: Multinomial Naïve Bayes (i.e., binary representation of the features) and Bernoulli Naïve Bayes (i.e., features are represented with frequency) [32]. Many studies have used NBCs for text, documents and products classification. A comparative study showed that NBC has higher accuracy to classify documents than other common classifiers, such as decision trees, neural networks, and support vector machines [38]. Collecting 7000 status updates (e.g., positive or negative) from 90 Facebook users, researchers found that NBC has a higher rate (77%) of accuracy to predict the sentimental status of users compared to the Rocchio Classifier (75%) [37]. Previous studies investigating different techniques of sentiment analysis [39] found that symbolic techniques (i.e., based on the force and direction of words) have accuracy lower than 80%. In contrast, machine learning techniques (e.g., SVM, NBC, and maximum Entropy) have a higher level of accuracy (above 80%) in classifying sentiment. NBCs can be used with limited size training data to estimate necessary parameters and are quite efficient to implement, as compared to other sophisticated methods with comparable accuracy [34]. However, NBCs are based on over-simplified assumptions of conditional probability and shape of data distribution [34,36].

2.3.3. Logistic Regression

Logistics regression (LR) is one of the popular and earlier methods for classification. LR was first developed by David Cox in 1958 [36]. In the LR model, the probabilities describing the possible outcomes of a single trial are modeled using a logistic function [34]. Using a logistic function, the probability of the outcomes are transformed into binary values (0 and 1). Maximum likelihood estimation methods are commonly used to minimize error in the model. A comparative study classifying product reviews reported that logistic regression multi-class classification method has the highest (min 32.43%, max 58.50%) accuracy compared to Naïve Bayes, Random Forest, Decision Tree, and Support Vector Machines classification methods [40]. Using multinomial logistic regression [41] observed that this method can accurately predict the sentiment of Twitter users up to 74%. Past research using stepwise logistic discriminant analysis [42] correctly classified 96.2% cases. LR classifier is suitable for predicting categorical outcomes. However, this prediction needs each data point to be independent to each other [36]. Moreover, the stability of the logistic regression classifier is lower than the other classifiers due to the widespread distribution of the values of average classification accuracy [40]. LR classifiers have a fairly expensive training phase which includes parameter modeling with optimization techniques [32].

### 2.3.4. K-Nearest Neighbor

K-Nearest Neighbor (KNN) is a popular non-parametric text classifier which uses instance-based learning (i.e., does not construct a general internal model but just stores an instance of the data) [34,36]. KNN method classifies texts or documents based on similarity measurement [32]. The similarity between two data points is measured by estimating distance, proximity or closeness function [43]. KNN classifier computes classification based on a simple majority vote of the nearest neighbors of each data point [34,44]. The number of nearest neighbors (K) is determined by specification or by estimating the number of neighbors within a fixed radius of each point. KNN classifiers are simple, easy to implement and applicable for multi-class problems [36,44,45].

### 2.3.5. Summary

Table 4 represents main features of different classifiers with their respective strengths and weaknesses. This table provides a good overview of all the classifiers mentioned in the above section. Based on a review of multiple machine learning methods, we decided to apply Naïve Bayes and logistic regression classification methods to train and test binary sentiment categories associated with the Coronavirus Tweets data. Naïve Bayes and logistic regression classification methods were selected based on their parsimony, and their proven performance with textual classification provides for interesting comparative evaluations.

**Table 4.** Summary of classifiers for machine learning.

| Classifier | Characteristic | Strength | Weakness |
|---|---|---|---|
| Linear regression | Minimize sum of squared differences between predicted and true values | Intuitive, useful and stable, easy to understand | Sensitive to outliers; Ineffective with non-linearity |
| Logistic regression | Probability of an outcome is based on a logistic function | Transparent and easy to understand; Regularized to avoid over-fitting | Expensive training phase; Assumption of linearity |
| Naïve Bayes classifier | Based on assumption of independence between predictor variables | Effective with real-world data; Efficient and can deal with dimensionality | Over-simplified assumptions; Limited by data scarcity |
| K-Nearest Neighbor | Computes classification based on weights of the nearest neighbors, instance based | KNN is easy to implement, efficient with small data, applicable for multi-class problems | Inefficient with big data; Sensitive to data quality; Noisy features degrade the performance |

## 3. Methods and Textual Data Analytics

The Methods section has two broad parts, the first part deals with exploratory textual analytics, summaries by features endogenous and exogenous to the text of the Tweets, data visualizations, and describes key characteristics of the Coronavirus Tweets data. It goes beyond traditional statistical summaries for quantitative and even ordinal and categorical data, because of the unique properties of textual data, and exploits the potential to fragment and synthesize textual data (such as by considering parts of the Tweets, "#" tags, assign sentiment scores, and evaluation of use of characters) into useful features which can provide valuable insights. This part of the analysis also develops textual analytics specific data visualizations to gain and present quick insights into the use of key words associated with Coronavirus and COVID-19. The second part deals with machine learning techniques for classification of textual data into positive and negative sentiment categories. Implicit therefore, is that the first part of the analytics also includes sentiment analysis of the textual component of Twitter data. Tweets are assigned sentiment scores using R and R packages. The Tweets with their sentiment scores, are then

split into train and test data, to apply machine learning classification methods using two prominent methods described below, and their results are discussed.

### 3.1. Exploratory Textual Analytics

Exploratory textual analytics deals with the generation of descriptors for textual features in data with textual variables, and the potential associations of such textual features with other non-textual variables in the data. For example, a simple feature that is often used in the analysis of Tweets is the number of characters in the Tweet, and this feature can also be substituted or augmented by measures such as the number of words per Tweet [9]. A "Word Cloud" is a common and visually appealing early stage textual data visualization, consisting of the size and visual emphasis of words being weighted by their frequency of occurrence in the textual corpus, and is used to portray prominent words in a textual corpus graphically [46]. Early stage World Clouds used plain vanilla black and white graphics, such as in Figure 2, and current representations use diverse word configurations (such as all word being set to horizontal orientation), colors and outline shapes, such as in Figure 4, for increased aesthetic impact. This research used R along with Wordcloud and Wordcloud2 packages, while other packages in R and Python are also available with unique Wordcloud plotting capabilities.

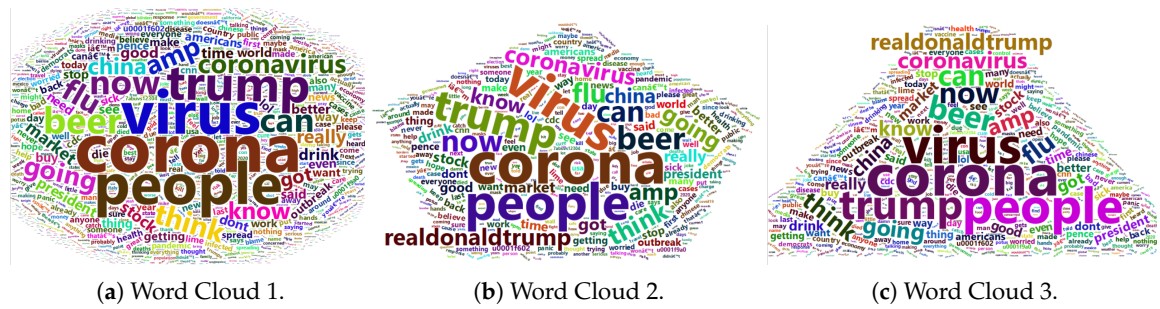

(**a**) Word Cloud 1.　　　　(**b**) Word Cloud 2.　　　　(**c**) Word Cloud 3.

**Figure 4.** A couple of word cloud instances.

### 3.1.1. Data Acquisition and Preparation

The research was initiated with standard and commonly used Tweets collection, cleaning and data preparation process, which we outline briefly below. We downloaded Tweets using a Twitter API, the rTweet package in R, which was used to gather over nine hundred thousand tweets from February to March of 2020, applying the keyword "Corona" (case ignored). This ensured a textual corpus focused on the Coronavirus, COVID-19 and associated phenomena, and reflects an established process for topical data acquisition [21,47]. The raw data with ninety variables was processed and prepared for analysis using the R programming language and related packages. The data was subset to focus on Tweets tagged by country as belonging to the United States. Multiple R packages were used in the cleaning process, to create a clean dataset for further analysis. Since the intent was to use the data for academic research, we replaced all identifiable abusive words with a unique alphanumeric tag word, which contained the text "abuvs", but was mixed with numbers to avoid using a set of characters that could have preexisted in the Tweets. Deleting abusive words completely would deprive the data of potential analyses opportunities, and hence a specifically coded algorithm was used to make a customized replacement. This customized replacement was in addition to the standard use of "Stopwords" and cleaning processes [48,49]. The dataset was further evaluated to identify the most useful variables, and sixty two variables with incomplete, blank and irrelevant values were deleted to create a cleaned dataset with twenty eight variables. The dataset was also further processed based on the needs of each analytical segment of analysis, using "tokenization"—which converts text to analysis relevant word tokens, "part-of-speech" tagging—which tags textual artifacts by grammatical category such as noun or verb, "parsing"—which identifies underlying structure between textual elements, "stemming"—which discards prefixes and suffixes using rules to create simple forms of base

words and "lemmatization"—which like stemming, aims to transform words to simpler forms and uses dictionaries and more complex rules and processes than in stemming.

### 3.1.2. Word and Phrase Associations

An important and distinct aspect of textual analytics involves the identification of not only the most frequently used words, but also of word pairs and word chains. This aspect, known as *N*-grams identification in a text corpus, has been developed and studied in computational linguistics and NLP. We transformed the "Tweets" variable, containing the text of the Tweets in the data, into a text corpus and identified the most frequent words, the most frequent Bigrams (two word sequences), the most frequent Trigrams (three word sequences) and the most frequent "Quadgrams" (four word sequences, also called Four-grams). Our research also explored longer sequences but the text corpus did not contain longer sequences with sufficient frequency threshold and relevance. While identification of *N*-grams is a straightforward process with the availability of numerous packages in R and Python, and other NLP tools, it is more nuanced to identify the most useful n-grams in a text corpus, and interpret the implications. In reference to Figure 5, it is seen that in some scenarios, such as with the popular use of words "beer", "Trump" and "abuvs" (the tag used to replace identifiable abusive words), and Bigrams and Trigrams such as "corona beer", "stock market", "drink corona", "corona virus outbreak", and "confirmed cases (of) corona virus" (a Quadgram) indicate a mixed mass response to the Coronavirus in its early stages. Humor, politics, and concerns about the stock market and health risks words were all mixed in early Tweets based public discussions on Coronavirus. Additional key word and sentiment analysis factoring the timeline, showed an increase in seriousness, and fear in particular as shown in Figure 1, indicating that public sentiment changed as the consequences of the rapid spread of Coronavirus, and the damaging impact upon COVID-19 patients became more evident.

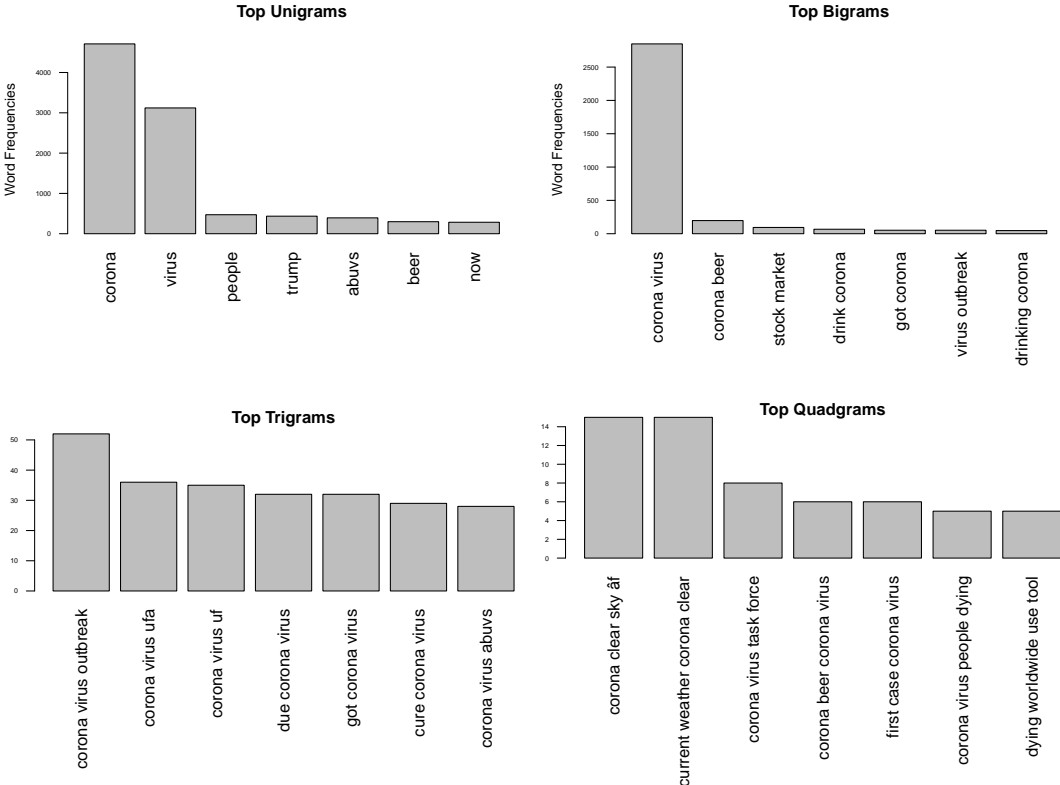

**Figure 5.** *N*-Grams.

### 3.1.3. Geo-Tagged Analytics

Data often contain information about geographic locations, such as city, county, state and country level data or by holding zip code and longitude and latitude coordinates, or geographical metadata. Such data are said to be "Geo-tagged", and "Geo-tagged Analytics" represents the analysis of data inclusive of geographical location variables or metadata. Twitter data contains two distinct location data types for each tweet: one is a location for the tweet, indicating where the Tweet was posted from, and the other is the general location of the user, and may refer to the place of stay for the user when the Twitter account was created, as shown in Table 5. For the Coronavirus Tweets, we examined both fear-sentiment and negative sentiment and found some counter-intuitive insights, showing relatively lower levels of fear in states which were significantly affected by a high number of COVID-19 cases, as demonstrated in Figure 3.

**Table 5.** Location variables (tagged and stated locations).

| Tagged | Frequency | Stated | Frequency |
|---|---|---|---|
| Los Angeles, CA | 183 | Los Angeles, CA | 78 |
| Manhattan, NY | 130 | United States | 75 |
| Florida, USA | 84 | Washington, DC | 60 |
| Chicago, IL | 71 | New York, NY | 54 |
| Houston, TX | 65 | California, USA | 49 |
| Texas, USA | 57 | Chicago, IL | 40 |
| Brooklyn, NY | 51 | Houston, TX | 39 |
| San Antonio, TX | 51 | Corona, CA | 33 |

### 3.1.4. Association with Non-Textual Variables

This research also analyzed Coronavirus Tweets texts for potential association with other variables, in addition to endogenous analytics, and the time and dates variable. Using a market segmentation logic, we grouped Tweets by the top three source devices in the data, namely: iPhone, Android and iPad, as shown in Figure 6, which is normalized to each device count. This means that Figure 6 reflects comparison of the relative ratio of device property count to total device count for each source category, and is not a direct device-totals comparison. Our research analyzed direct totals comparison as well, and the reason for presenting the source device comparison by relative ratio is because the comparison by totals simply follows the distribution of source device totals provided in Table 1. We observed that, higher ratio of: iPhone users made the most use of hashtags and mentions of "Corona", iPad users made the most mention of URLs and "Trump", Android users made the most mention of "Flu" and "Beer" words. Both iPhone and Android users has similar ratios for usage of abusive words.

### 3.1.5. Sentiment Analytics

One of the key insights that can be gained from textual analytics is the identification of sentiment associated with the text being analyzed. Extant research has used custom methods to identify temporal sentiment as well as sentiment expressions of character traits such as dominance [17], and standardized methods to assign positive and negative sentiment scores [7,50]. Sentiment analysis is broadly described as the assignment of sentiment scores and categories, based on keyword and phrase match with sentiment score dictionaries, and customized lexicons. Prominent analytics software including R, and open-source option, have standardized sentiment scoring mechanisms. We used two R packages, Syuzhet and sentimentr, to classify and score the Tweets for sentiment classes such as fear, sadness and anger, and sentiment scores ranging from negative (around $-1$) to positive (around 1) with sentiR [51,52]. We used two methods to assign sentiment scores and classifications: the first method assigned a positive to negative score as continuous value between 1 (maximum positive) and $-1$ (minimum positive).

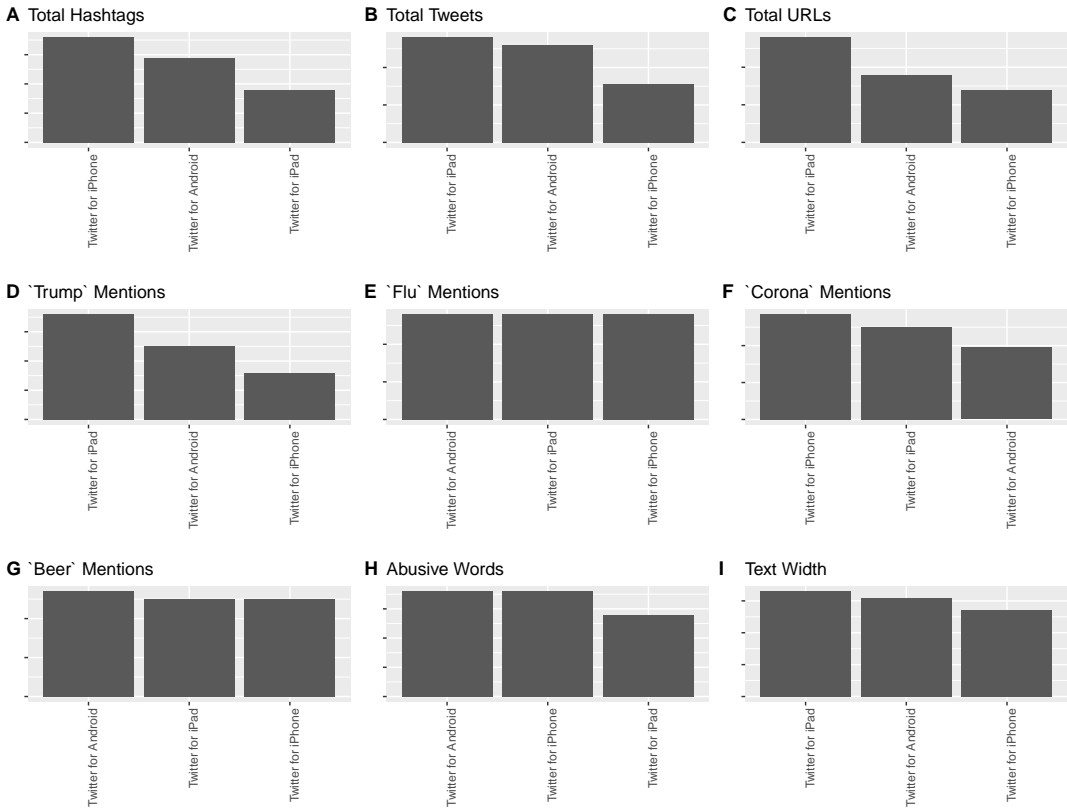

**Figure 6.** Source device comparison by relative ratio.

### 3.2. Machine Learning with Classification Methods

Extant research has examined linguistic challenges and has demonstrated the effectiveness of ML methods such as SVM (Support Vector Machine) in identifying extreme sentiment [53]. The focus of this study is on demonstrating how commonly used ML methods can be applied, and used to contribute to classification of sentiment by varying Tweets characteristics, and not the development of contributions to new ML theory or algorithms. Unlike linear regression, which is mainly used for estimating the probability of quantitative parameters, classification can be effectively used for estimating the probability of qualitative parameters for binary or multi-class variables—that is when the prediction variable of interest is binary, categorical or ordinal in nature. There are many classification methods (classifiers) for qualitative data; among the most well-known are Naïve Bayes, logistic regression, linear and KNN. The first two are elaborated upon below in the context of textual analytics. The most general form of classifiers is as follows:

How can we predict responses $Y$ given a set of predictors $\{X\}$? For general linear regression, the mathematical model is $Y = \beta_0 + \beta_1 x_1 + \beta_2 x_2 +, \cdots, + \beta_n x_n$. The aim is to find an estimated $\widehat{Y}$ for $Y$ by modeling values of $\hat{\beta}_0, \hat{\beta}_1, \cdots, \hat{\beta}_n$ for $\beta_0, \beta_1, \cdots, \beta_n$. These estimates are determined from training data sets. If either the predictors and/or responses are not continuous quantitative variables, then the structure of this model is inappropriate and needs modifications. $X$ and $Y$ become proxy variables and their meaning depends on the context in which they are used; in the context of the present study, $X$ represents a document or features of a document and $Y$ is the class to be evaluated, for which the model is being trained.

Below is a brief mathematical-statistical formulation of two of the most important classifiers for textual analytics, and sentiment classification in particular: Naïve Bayes which is considered to be a generative classifier, and Logistic Regression which is considered to be a discriminative classifier. Extant research has demonstrated the viability of the using Naïve Bayes and Logistic Regression for generative and discriminative classification, respectively [54].

*3.3. Naïve Bayes Classifier*

Naïve Bayes Classifier is based on Bayes conditional probability rule [55]. According to Bayes theorem, the conditional probability of $P(x|y)$ is,

$$P(x|y) = \frac{P(y|x)P(x)}{P(y)} \tag{1}$$

The Naïve Bayes classifier identifies the estimated class $\hat{c}$ among all the classes $c \in C$ for a given document $d$. Hence the estimated class is,

$$\hat{c} = argmax_{c \in C} P(c|d) \tag{2}$$

After applying Bayes conditional probability from (1) in (2) we get,

$$\hat{c} = argmax_{c \in C} P(c|d) = argmax_{c \in C} \frac{P(d|c)P(c)}{P(d)} \tag{3}$$

Simplifying (3) (as $P(d)$ is the same for all classes, we can drop $P(d)$ from the denominator) and using the likelihood of $P(d|c)$, we get,

$$\hat{y} = argmax_{c \in C} P(y_1, y_2, \cdots, y_n|c)P(c) \tag{4}$$

where $y_1, y_2, \cdots, y_n$ are the representative features of document $d$.

However, (4) is difficult to evaluate and needs more simplification. We assume that word position does not have any effect on the classification and the probabilities $P(y_i|c)$ are independent given a class $c$, hence we can write,

$$P(y_1, y_2, \cdots, y_n|c) = P(y_1|c).P(y_2|c).\cdots.P(y_n) \tag{5}$$

Hence, from (4) and (5) we get the final equation of the Näive Bayes classifier as,

$$C_{NB} = argmax_{c \in C} P(c) \prod_{y_i \in Y} P(y_i|c) \tag{6}$$

To apply the classifier in the textual analytics, we consider the index position of words $(\mathbf{w}_i)$ in the documents, namely replace $y_i$ by $\mathbf{w}_i$. Now considering features in log space, (6) becomes,

$$C_{NB} = argmax_{c \in C} logP(c) + \sum_{i \in positions} logP(\mathbf{w}_i|c) \tag{7}$$

3.3.1. Classifier Training

In (7), we need to find the values of $P(c)$ and $P(\mathbf{w}_i|c)$. Assume $N_c$ and $N_{doc}$ denote the number of documents in the training data belong in class $c$ and the total number of documents, respectively. Then,

$$\hat{P}(c) = \frac{N_c}{N_{doc}} \tag{8}$$

The probability of word $\mathbf{w}_i$ in class $c$ is,

$$\hat{P}(w_i|c) = \frac{count(\mathbf{w}_i, c)}{\sum_{\mathbf{w} \in V} count(\mathbf{w}, c)} \tag{9}$$

where $count(\mathbf{w}_i, c)$ is the number of occurrences of $\mathbf{w}_i$ in class $c$, and $V$ is the entire word vocabulary.

Now since Naïve Bayes multiplies all the features likelihood together (refer to (6)), the zero probabilities in the likelihood term for any class will turn the whole probability to zero, to avoid such situation, we use the Laplace add-one smoothing method, hence (9) becomes,

$$\hat{P}(w_i|c) = \frac{count(\mathbf{w}_i, c) + 1}{\sum_{\mathbf{w} \in V} (count(\mathbf{w}, c) + 1)}$$
$$= \frac{count(\mathbf{w}_i, c) + 1}{\sum_{\mathbf{w} \in V} (count(\mathbf{w}, c) + |V|)}$$

(10)

From an applied perspective, the text needs to be cleaned and prepared to contain clear, distinct and legitimate words ($\mathbf{w}_i$) for effective classification. Custom abbreviations, spelling errors, emoticons, extensive use of punctuation, and such other stylistic issues in the text can impact the accuracy of classification in both the Naïve Bayes and logistic classification methods, as text cleaning processes may not be 100% successful.

### 3.4. Application of Naïve Bayes for Coronavirus Tweet Classification

This research aims to explore the viability of applying exploratory sentiment classification in the context of Coronavirus Tweets. The goal therefore was directional, and set to classifying positive sentiment and negative sentiment in Coronavirus Tweets. Tweets with positive sentiment were assigned a value of 1, and Tweets with a negative sentiment were assigned a value of 0. We created subsets of data based on the length of Tweets to examine classification accuracy based on length of Tweets, where the lengths of Tweets were calculated by a simple character count for each Tweet. We created two groups, where the first group consisted of Coronavirus Tweets which were less than 77 characters in length, consisting of about a quarter of all Tweets data, and the group consisted of Coronavirus Tweets which were less than 120 characters in length, consisting of about half of all Tweets data. These groups of data were further subset to ensure that the number of positive Tweets and Negative Tweets were balanced when being classified. We used R [56] and associated packages to run the analysis, train using a subset of the data, and test the accuracy of the classification method using about 70 randomized test values. The results of using Naïve Bayes for Coronavirus Tweet Classification are presented in Table 6.

**Table 6.** Naïve Bayes classification by varying Tweet lengths.

| | Tweets (nchar < 77) | | | Tweets (nchar < 120) | |
| --- | --- | --- | --- | --- | --- |
| | **Negative** | **Positive** | | **Negative** | **Positive** |
| Negative | 34 | 1 | Negative | 34 | 1 |
| Positive | 5 | 30 | Positive | 29 | 6 |
| | Accuracy: 0.9143 | | | Accuracy: 0.5714 | |

Interestingly, though we found strong classification accuracy for shorter Tweets with around nine out of every ten Tweets being classified correctly (91.43% accuracy). We observed an inverse relationship between the length of Tweets and classification accuracy, as the classification accuracy decreased to 57% with increase in the length of Tweets to below 120 characters. We calculated the Sensitivity of the classification test, which is given by the ratio of the number of correct positive predictions (30) in the output, to the total number of positives (35), to be 0.86 for the short Tweets and 0.17 for the longer Tweets. We calculated the Specificity of the classification test, which is given by the ratio of the number of correct negative predictions (34) in the output, to the total number of negatives (35), to be 0.97 for both the short and long Tweets classification. Naïve Bayes thus had better performance with classifying negative Tweets.

*3.5. Logistic Regression*

Logistic regression is a probabilistic classification method that can be used for supervised machine learning. For classification, a machine learning model usually consists of the following components [54]:

1.  **A feature representation of the input:** For each input observation $(x^{(i)})$, this will be represented by a vector of features, $[x_1, x_2, \cdots, x_n]$.
2.  **A classification function**: It computes the estimated class $\hat{y}$. The sigmoid function is used in classification.
3.  **An objective function:** The job of objective function is to minimize the error of training examples. The cross-entropy loss function is often used for this purpose.
4.  **An optimizing algorithm:** This algorithm will be used for optimizing the objective function. The stochastic gradient descent algorithm is popularly used for this task.

3.5.1. The Classification Function

Here we use logistic regression and sigmoid function to build a binary classifier.

Consider an input observation $x$ which is denoted by a vector of features $[x_1, x_2, \cdots, x_n]$. The output of classifier will be either $y = 1$ or $y = 0$. The objective of the classifier is to know $P(y = 1|x)$, which denotes the probability of positive sentiment in this classification of Coronavirus Tweets, and $P(y = 0|x)$, which correspondingly denotes the probability of negative sentiment. $w_i$ denotes the weight of input feature $x_i$ from a training set and $b$ denotes the bias term (intercept), we get the resulting weighted sum for a class,

$$z = \sum_{i=1}^{n} w_i.x_i + b \tag{11}$$

representing $w.x$ as the element-wise dot product of vectors of $w$ and $x$, we can simplify (11) as,

$$z = w.x + b \tag{12}$$

We use the following sigmoid function to map the real-valued number into the range $[0, 1]$,

$$y = \sigma(z) = \frac{1}{1 + e^{-z}} \tag{13}$$

After applying sigmoid function in (12) and making sure that $P(y = 1|x) + P(y = 0|x) = 1$, we get the following two probabilities,

$$P(y = 1|x) = \sigma(w.x + b)$$
$$= \frac{1}{1 + e^{-(w.x+b)}} \tag{14}$$

$$P(y = 0|x) = 1 - P(y = 1|x)$$
$$= \frac{e^{-(w.x+b)}}{1 + e^{-(w.x+b)}} \tag{15}$$

considering 0.5 as the decision boundary, the estimated class $\hat{y}$ will be,

$$\hat{y} = \begin{cases} 1 & \text{if } P(y = 1|x) > 0.5 \\ 0 & \text{otherwise} \end{cases} \tag{16}$$

### 3.5.2. Objective Function

For an observation $x$, the loss function computes how close the estimated output $\hat{y}$ is from the actual output $y$, which is represented by $L(\hat{y}, y)$. Since there are only two discrete outcomes ($y = 1$ or $y = 0$), using Bernoulli distribution, $P(y|x)$ can be expressed as,

$$P(y|x) = \hat{y}^y (1 - \hat{y})^{1-y} \tag{17}$$

taking log both sides in (17),

$$\begin{aligned} \log P(y|x) &= \log\left[\hat{y}^y (1 - \hat{y})^{1-y}\right] \\ &= y \log \hat{y} + (1 - y) \log(1 - \hat{y}) \end{aligned} \tag{18}$$

To turn (18) into a minimizing function (loss function), we take the negation of (18), which yields,

$$L(\hat{y}, y) = -\left[y \log \hat{y} + (1 - y) \log(1 - \hat{y})\right] \tag{19}$$

substituting $\hat{y} = \sigma(w.x + b)$, from (19), we get,

$$\begin{aligned} L(\hat{y}, y) &= -\left[y \log \sigma(w.x + b) + (1 - y) \log(1 - \sigma(w.x + b))\right] \\ &= -\left[y \log\left(\frac{1}{1 + e^{-(w.x+b)}}\right) + (1 - y) \log\left(\frac{e^{-(w.x+b)}}{1 + e^{-(w.x+b)}}\right)\right] \end{aligned} \tag{20}$$

### 3.5.3. Optimization Algorithm

To minimize the loss function stated in (20), we use gradient descent method. The objective is to find the minimum weight of the loss function. Using gradient descent, the weight of the next iteration can be stated as,

$$w^{k+1} = w^k - \eta \frac{\mathrm{d}}{\mathrm{d}w} f(x; w) \tag{21}$$

where $\frac{\mathrm{d}}{\mathrm{d}w} f(x; w)$ is the slope and $\eta$ is the learning rate.

Considering $\theta$ as vector of weights and $f(x; \theta)$ representing $\hat{y}$, the updating equation using gradient descent is,

$$\theta^{k+1} = \theta^k - \eta \nabla_\theta L\left(f(x; \theta), y\right) \tag{22}$$

where

$$\begin{aligned} L\left(f(x; \theta), y\right) &= L(w, b) \\ &= L(\hat{y}, y) = -\left[y \log \sigma(w.x + b) + (1 - y) \log(1 - \sigma(w.x + b))\right] \end{aligned} \tag{23}$$

and the partial derivative ($\frac{\partial}{\partial w_j}$) for this function for one observation vector $x$ is,

$$\frac{\partial L(w, b)}{\partial w_j} = \left[\sigma(w.x + b) - y\right] x_j \tag{24}$$

where the gradient in (24) represents the difference between $\hat{y}$ and $y$ multiplied by the corresponding input $x_j$. Please note that in (22), we need to do the partial derivatives for all the values of $x_j$ where $1 \leq j \leq n$.

### 3.6. Application of Logistic Regression for Coronavirus Tweet Classification

As described in Section 3.4, the purpose is to demonstrate application of exploratory sentiment classification, to compare the effectiveness of Naïve Bayes and logistic regression, and to examine

accuracy under varying lengths of Coronavirus Tweets. As with classification of Tweets using Naïve Bayes, positive sentiment Tweets were assigned a value of 1, and negative sentiment Tweets were denoted by 0, allowing for a simple binary classification using logistic regression methodology. Subsets of data were created, based on the length of Tweets, in a similar process as for Naïve Bayes classification and the same two groups of data containing Tweets with less than 77 characters (approximately 25% of the Tweets), and Tweets with less than 125 characters (approximately 50% of the data) respectively, were used. We used R [56] and associated packages for logistic regression modeling, and to train and test the data. The results of using logistic regression for Coronavirus Tweet Classification are presented in Table 7.

**Table 7.** Logistic classification by varying Tweet lengths.

| | Tweets (nchar < 77) | | | Tweets (nchar < 120) | |
|---|---|---|---|---|---|
| | **Negative** | **Positive** | | **Negative** | **Positive** |
| Negative | 30 | 5 | Negative | 21 | 14 |
| Positive | 13 | 22 | Positive | 19 | 16 |
| | Accuracy: 0.7429 | | | Accuracy: 0.52 | |

We observed on the test data with 70 items that, akin to the Naïve Bayes classification accuracy, shorter Tweets were classified using logistic regression with a greater degree of accuracy of just above 74%, and the classification accuracy decreased to 52% with longer Tweets. We calculated the Sensitivity of the classification test, which is given by the ratio of the number of correct positive predictions (22) in the output, to the total number of positives (35), to be 0.63 for the short Tweets, and 0.46 for the longer Tweets. We calculated the Specificity of the classification test, which is given by the ratio of the number of correct negative predictions (30) in the output, to the total number of negatives (35), to be 0.86 for the short Tweets, and 0.60 for the longer Tweets classification. Logistic regression thus had better performance with a balanced classification of Tweets.

## 4. Discussion

The classification results obtained in this study are interesting and indicate a need for additional validation and empirical model development with more Coronavirus data, and additional methods. Models thus developed with additional data and methods, and using Naïve Bayes and logistic regression Tweet Classification methods can then be used as independent mechanisms for automated classification of Coronavirus sentiment. The model and the findings can also be further extended to similar local and global pandemic insights generation in the future. Textual analytics has gained significant attention over the past few years with the advent of big data analytics, unstructured data analysis and increased computational capabilities at decreasing costs, which enables the analysis of large textual datasets. Our research demonstrates the use of the NRC sentiment lexicon, using the Syuzhet and sentimentr packages in R ([51,52]), and it will be a useful exercise to evaluate comparatively with other sentiment lexicons such as Bing and Afinn lexicons [51]. Furthermore, each type of text corpus will have its own features and peculiarities, such as Twitter data will tend to be different from LinkedIn data in syntactic features and semantics. Past research has also indicated the usefulness of applying multiple lexicons, to generate either a manually weighted model or a statistically derived model based on a combination of multiple sentiment scores applied to the same text, and hybrid approaches [57], and a need to apply strategic modeling to address big data challenges. We have demonstrated a structured approach which is necessary for successful generation of insights from textual data. When analyzing crisis situations, it is important to map sentiment against time, such as in the fear curve plot (Figure 1), and where relevant geographically, such as in Figure 3a,b. Associating text and textual features with carefully selected and relevant non-textual features is another critical aspect of insights generation through textual analytics as has been demonstrated through Tables 1–7.

### 4.1. Limitations

The current study focused on a textual corpus consisting of Tweets filtered by "Coronavirus" as the keyword. Therefore the analysis and the methods are specifically applied to data about a particular pandemic as a crisis situation, and hence it could be argued that the analytical structure outlined in this paper can only be weakly generalized. Future research could address this and explore "alternative dimensionalities and perform sensitivity analysis" to improve the validity of the insights gained [58]. The Novel Coronavirus pandemic is a phenomena of an unprecedented nature, and associated social media trends can also therefore be considered to possess distinct characteristics. Hence, it is important to contextualize the use of ML, because using data from pre-COVID-19 time periods mixed with one of more phases of the spread of the virus would confound ML modeling and results, unless control mechanisms are used to control for pandemic effects. The present study addresses this data-validity challenge in applying machine learning by using Twitter data, filtered and processed to provide a clean Coronavirus dataset, from a single phase of the spread of the pandemic. ML was applied to classify sentiment for Tweets only within this period, and is therefore justifiably useful for classifying COVID-19 Tweets sentiment. The data-validity limitation for unique events such as COVID-19 must be accounted for in future studies using machine learning on pandemic associated data.

Furthermore, the analysis used one sentiment lexicon to identify positive and negative sentiments, and one sentiment lexicon to classify the tweets into categories such as fear, sadness, anger and disgust [7,51,52]. Varying information categories have the potential to influence human beliefs and decision making [59], and hence it is important to consider multiple social media platforms with differing information formats (such as short text, blogs, images and comments) to gain a holistic perspective. The present study intended to generate rapid insights for COVID-19 related public sentiment using Twitter data, which was successfully accomplished. This study also intended to explore the viability of machine learning classification methods, and we found sufficient directional support for the use of Naïve Bayes and Logistic classification for short to medium length Tweets, but the accuracy decreased with the increase in the length of Tweets. We have not stated a formal model for Tweets sentiment classification, as that is not a goal of this research. While the absence of such a formal model may also be perceived as a limitation which we acknowledge, it must be noted that our research goal of evaluating the viability of using machine learning classification for Tweets of varying lengths was accomplished. Finally, we also acknowledge that Twitter data alone is not a reflection of general mass sentiment in a nation or even in a state or local area [8,11,29]. However, the current research provides a clear direction for more comprehensive analysis of multiple textual data sources including other social media platforms, news articles and personal communications data. The mismatch between Coronavirus negative sentiment map, fear sentiment map, and the factually known hot spots in New York, New Jersey and California, as shown in Figure 3 could have been driven by the timing of tweets posted just before the magnitude of the problem was recognized, and could also be reflective of cultural attitudes. The sentiment map presents a fair degree of acceptable association with states such as West Virginia and North Dakota. Overall, though these limitations are acknowledged from a general perspective, they do not diminish the contributions made by this study, as the generic weaknesses are not associated with the primary goals of this study.

### 4.2. Implications and Ethics

There have been some ethical concerns about the way in Twitter data has been used for research and by practitioners—numerous potential issues were identified, including the use of Tweets made by vulnerable persons in crisis situations [60]. It is also important to recognize the deviation from researcher obligations to human subjects, to researcher obligations to "data subjects" [61], and this approach does not compromise on ethics, but rather acknowledges the value of publicly available data as voluntary contributions to public space by Twitter users. Past research also identified the use of Twitter data analytics for pandemics, including the 2009 Swine Flu [60], indicating a mature

stream of thought towards using social media data to help understand and manage contagions and crisis scenarios.

As a global pandemic COVID-19 is adversely affecting people and countries. Besides necessary healthcare and medical treatments, it is critical to protect people and societies from psychological shocks (e.g., distress, anxiety, fear, mental illness). In this context, automated machine learning driven sentiment analysis could help health professionals, policymakers, and state and federal governments to understand and identify rapidly changing psychological risks in the population. Consequently, timely responses and initiatives (e.g., counseling, internet-based psychological support mechanisms) taken by the agencies to mitigate and prevent adverse emotional and psychological consequences will significantly improve public health and well being during crisis phenomena. Sentiment analysis using social media data will thus provide valuable insights on attitudes, perceptions, and behaviors for critical decision making for business and political leaders, and societal representatives.

## 5. Conclusions and Future Work

We addressed issues surrounding public sentiment reflecting deep concerns about Coronavirus and COVID-19, leading to the identification of growth in fear sentiment and negative sentiment. We also demonstrated the use of exploratory and descriptive textual analytics and textual data visualization methods, to discover early stage insights, such as by grouping of words by levels of a specific non-text variable. Finally, we provided a comparison of textual classification mechanisms used in artificial intelligence applications, and demonstrated their usefulness for varying lengths of Tweets. Thus, the present study presented methods with valuable informational and public sentiment insights generation potential, which can be used to develop much needed motivational solutions and strategies to counter the rapid spread of "the trio of fear-panic-despair" associated with Coronavirus and COVID-19 [13]. Given the easy availability of COVID-19 related big data, an extensive array of analytics and state of the art machine learning driven solutions needs to be developed to address the pandemic's global information complexities. While the current research stream contributes to the strategic process, a lot more needs to be done across multiple social media, news and public and personal communication platforms. Such solutions will also be critical in identifying a sustainable pathway to recovery post-COVID-19: for example, understanding public perspectives and sentiment using textual analytics and machine learning will enable policy makers to cater to public needs more specifically and also design sentiment specific communication strategies. Corporations and small businesses can also benefit through such analyses and machine learning models to better understand consumer sentiment and expectations. Our research is ongoing, and we are building on the foundations laid in this paper to analyze larger volumes of new data which are expected to help build models to support the socioeconomic recovery process in the time ahead.

**Author Contributions:** This work was completed with contributions from all the authors. conceptualization, J.S., G.G.M.N.A., Y.S. and M.M.R.; methodology, J.S. and M.M.R.; software, J.S.; validation, J.S., M.M.R., E.E. and Y.S.; formal analysis, J.S. and M.M.R.; data curation, J.S.; writing—original draft preparation, J.S., G.G.M.N.A., M.M.R., E.K., and Y.S.; writing—review and editing, J.S., G.G.M.N.A., M.M.R., E.E., and Y.S.; visualization, J.S., and M.M.R.; funding acquisition, G.G.M.N.A. All authors did edit, review and improve the manuscript. All authors have read and agreed to the published version of the manuscript.

**Funding:** This research did not receive any external funding.

**Conflicts of Interest:** The authors declare no conflict of interest.

## Abbreviations

The following abbreviations are used in this manuscript:

| | |
|---|---|
| COVID-19 | Coronavirus Disease 2019 |
| ML | Machine Learning |
| NLP | Natural Language Processing |

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
