# Peer review of "COVID-19 Public Sentiment Insights and Machine Learning for Tweets Classification"

_information, doi:10.3390/info11060314_

Round 1

Reviewer 1 Report

- Some abbreviations must be defined before using, such as:
CSAIL, MIT (line 33)

- Figures should be clarified with more explanation for their sources.

- The authors should compare their experiments with recent machine learning approaches, such as deep learning, and focus more on linguistic features such as word embeddings.
Several studies that have compared the machine learning approaches and linguistic features may be as useful, such as:

Almatarneh, S., & Gamallo, P. (2019). Comparing supervised machine learning strategies and linguistic features to search for very negative opinions. Information, 10(1), 16.

- There are a lot of self-citation in this study

Reviewer 2 Report

This paper attempts to get insights from sentiment analysis of COVID-19 using Twitter data and classical tools such as Naive Bayes Classifier (NBC).

First of all, very verbose expressions stand out in the manuscript. The 21-page paper is very long. Since it is not a book or commentary paper, authors should discuss the works at an early stage, rather than writing long explanations of the old methods such as NBC, KNN, or LR. These are the worn-out techniques and do not require detailed explanations or mathematical formulas. Explain to us about your own research parts rather than introducing other researchers' works.

I think this manuscript would fit on 12 pages. The followings are the comments. Please read carefully and modify your paper.

  1. Figures 1 and 2 seem unnecessary. Figure 1 does not specify the vertical axis and the data source. Also, the meaning of the dotted line is not explained.
  2. Why did you use the word cloud? It is an excellent method to grasp the whole keyword intuitively, but there is no reason to be suitable in this experiment. Why not show the frequency of words in a bar graph or use a word co-occurrence network?
  3. The 69th line says, "Early stage exploratory analytics of Tweets revealed ..." but I do not know what the source is. 
  4. "[15]." on the 90th line, and ")" at the end of the 94th line are probably typos.
  5. Does "Frequency" in Table 2 represent the number of appearances of mentions? It may represent the top eight appearances, but it is not possible to judge because there is no explanation. Also, it seems very small compared to the number of tweets.
  6. Does Fig.3 need to show in the manuscript?
  7. Figures 4 and 5 are redundant. Instead of explaining the primary processing of NLP step by step, the authors should focus on their original approach.
  8. Most of the explanations for NBC in 3.2 and 3.3 and the part of LR in 3.5 can be omitted. The authors can remove the formulas except for the original part of this paper. Moreover, NBC, KNN, and LR are too classic to call machine learning.
  9. The authors evaluated their approach by the classification problem. However, I could not understand the classification accuracies meant. What kinds of problems will be solved by the improvement of accuracy?

I have to say that the manuscript is redundant as a whole. Although the content is well written, and it may be the reading material for beginners, it is hard to say that the discussion has reached the quality of the journal.

It is difficult to accept it, and the manuscript requires significant revision.

Reviewer 3 Report

The paper presents a great deal of work: comprehensive review of research in the field of Twitter data analysis and text analysis, review of methods and tools for text analysis, comparative analysis of machine learning procedures with application in text analysis, positive and negative sentiment analysis. The results are clearly presented and interpreted.

Corrective suggestions:

Please check that tables and figures appear after mentioning in text, e.g. fig. 4 appears before text reference; table 4 - the same problem

Line 90: after reference [15] delete point

Line 300: correct Cornovirus

Line 313: correct Cornonavirus

Round 2

Reviewer 1 Report

All comments have been handled

Author Response

We thank the reviewer for accepting our work.  

Reviewer 2 Report

This paper's approach may have a value as trial sharing, in the situation of COVID-19 spreading around the world. Although the quality of the analysis and discussion is still low, it is worth praising that the authors answered all the comments. However, it is generally desirable for the paper to be written concisely, and this paper is scattered with redundant expressions. As I commented, 22 pages are too long.
After all, the approach of this research must be questioned. COVID-19 turmoil has been described as a pandemic and infodemic that has never occurred in history. In other words, the current situation is that there is hardly enough data and sufficient experience to evaluate the learning results using ML. What does it mean to use ML methods on twitter data in this paper? The author may need to clarify that point. The readers may raise the question that the paper is only to use readily available ML methods and data to the topic in which people are interested.
